# Combination of Bone-Modifying Agents with Immunotarget Therapy for Hepatocellular Carcinoma with Bone Metastases

**DOI:** 10.3390/jcm11236901

**Published:** 2022-11-23

**Authors:** Zhaoyu Chen, Zhilong Shen, Xiang Wang, Pengru Wang, Xiaofei Zhu, Jiefu Fan, Bo Li, Wei Xu, Jianru Xiao

**Affiliations:** 1Department of Orthopedic Oncology, Changzheng Hospital, Naval Medical University (Second Military Medical University), 415 Fengyang Road, Shanghai 200003, China; 2Department of Radiation Oncology, Changhai Hospital, Naval Medical University (Second Military Medical University), 168 Changhai Road, Shanghai 200082, China; 3Department of Vascular Surgery, Changhai Hospital, Naval Medical University (Second Military Medical University), 168 Changhai Road, Shanghai 200082, China

**Keywords:** bone metastases, denosumab, immunotarget therapy, hepatocellular carcinoma, PD-1

## Abstract

Due to limited investigations about efficacy of tyrosine kinase inhibitors (TKIs) plus immune-checkpoint inhibitors (ICIs) versus TKIs alone, and effects of durations of bone modifying agents (BMAs) on the survival of patients with hepatocellular carcinoma (HCC) and bone metastases (BoM), we aim to compare the efficacy of TKIs both alone and in combination with ICIs, as well as comparing long-term and no or perioperative use of BMAs for patients with HCC and BoM. Patients with pathologically confirmed HCC and BoM were included in the study. They were stratified into the TKIs group and the TKIs + ICIs group, and the perioperative and the long-term use of BMAs group. Overall survival (OS), progression-free survival (PFS), objective response rate (ORR), and disease control rate (DCR) were calculated to assess the response to these regimes. The cumulative risk of initial skeletal-related events (SREs) was used to evaluate treatment efficacy for bone lesions. A total of 21 (33.9%) patients received TKIs (Sorafenib or Lenvatinib) alone and 41 (66.1%) received TKIs + ICIs. The combination group showed higher ORR than monotherapy group (1/21, 4.7% vs. 9/41, 22.0%; *p* = 0.1432); Additionally, the TKIs + ICIs group offered improved OS (18 months vs. 31 months; *p* = 0.015) and PFS (10 months vs. 23 months; *p* = 0.014), while this survival benefits were more profound in virus-infected patients than those non-infected. Prolonged OS (33 months vs. 16 months; *p* = 0.0048) and PFS (33 months vs. 11 months; *p* = 0.0027) were observed in patients with long-term use of BMAs compared with no or perioperative use of BMAs. The TKIs + ICIs combination and long-term adjuvant of BMAs may offer a survival advantage for HCC patients with BoM without severe adverse events, which requires further validations.

## 1. Background

Hepatocellular carcinoma (HCC), which accounts for 90% of liver cancers, has already become the second cause of cancer-related deaths, with an increasing incidence globally [1]. Due to patients’ gradually improved survival attributable to the development of multimodality therapy, the risk of bone metastases (BoM) is increasing to 25.5–38.5% [2]. The following skeletal-related events (SREs), such as metastatic bone pain, pathological fractures, and epidural spinal cord compressions, are independently relevant to worse clinical outcomes [3]. Albeit much emphasis has been placed on the BoM of lung, breast, and prostate cancer owing to its high incidence, clinical investigations on the BoM of HCC are limited.

The advent of immune-checkpoint inhibitors (ICIs) has changed the treatment paradigms of HCC, especially for patients with advanced stages. Given that monotherapy of tyrosine kinase inhibitors (TKIs) like Sorafenib or Lenvatinib has become the mainstay therapy of HCC [4,5] with the coming emergence and renovation of kinase inhibitors in recent years [6]. The combination therapy of TKIs + ICIs (targeted anti-PD-1/PD-L1), like pembrolizumab plus lenvatinib, has been verified as superior for prolonging the overall survival (OS) and progression-free survival (PFS) [7]. Recent retrospective studies have proved the safety and effectiveness of the combined regimen for advanced HCC [8,9]. However, there was a paucity of clinical research concentrating on the TKIs + ICIs, especially for HCC with BoM, while this regimen has been widely used as the first-line systemic therapy [10].

Bone-modifying agents (BMAs), zoledronic acid (ZA) and denosumab (the monoclonal antibody directed against RANKL) used for BoM patients has been commonly applied to restrain cancer-related pain, the progression of bone focus zone, and ensuing SREs. They are shown effective as the adjuvant therapy of solid tumors, including prostatic cancer (PC), melanoma and non-small cell lung cancer (NSCLC) [11,12]. Recent evidence has demonstrated that anti-RANKL can enhance the effect of ICIs against solid tumors and metastases [13,14]. A combination of BMAs (whether ZA or denosumab) and ICI/TKIs can improve clinical outcomes [12,15,16,17]. Nevertheless, no research has explored the response of anti-RANKL combined with TKIs + ICIs for HCC patients with BoM.

Therefore, we intended to compare the efficacy and safety of TKIs with TKIs plus ICIs for HCC with BoM. Meanwhile, an investigation was directed toward the potential effect of BMAs in the context of systemic therapy.

## 2. Methods

### 2.1. Study Design and Patients

This is a real-world single-center study by Changzheng Hospital, Shanghai, China. All patients between Jan. 2018 and Aug. 2021 were diagnosed with HCC with BoM by pathological findings and clinical symptoms, in terms of the criterion enacted by the American Association for the Study of Liver Diseases (AASLD) [18]. Bone metastases should be verified by bone lesion specimens evaluated via biopsies or postoperative pathology examinations. Waivers of our study were approved by the Ethics Committee of Changzheng Hospital (Ethics approved No. 2022SLYS5, 1 July 2022). The patients’ data were gathered from the medical records and imaging assessment performed during the perioperative and follow-up period, which has been consented to by the patients. When contacting the patient’s family members via telephone or other forms to obtain long-term treatment information, the consent of relatives is necessary.

### 2.2. Study Treatment and Basis for Grouping

This study included the patients with metastatic HCC and BoM who received a TKI (Sorafenib or Lenvatinib) as the first or second-line treatment and had the bone lesions resected in Changzheng Hospital. If the combination therapy of TKIs plus ICIs (Camrelizumab or Sintilimab) had not been used for more than three months, the patients would be stratified into the monotherapy group. Therefore, participants were classified into the TKIs only group and TKIs plus ICIs combination therapy group. Long-term use of BMAs was defined as the administration of more than three months. Furthermore, outcomes of perioperative use of BMAs and long-term use of BMAs were also evaluated.

### 2.3. Data Collection and Statistics

The following data was collected and analyzed: For general information, age, gender and Eastern Oncology Collaborative Group performance status score (ECOG-PS) have been collected. For preoperational data, Child–Pugh classification, AFP level, bone metastasis locations, HBV or HCV infection, Tomita score, Frankel score and whether surgery has been performed on the primary lesions have been collected. For sequential data, we collected whether radiotherapy or chemotherapy has been performed, visceral metastases, administrations of BMAs (type and duration), systemic therapy and adverse events (AEs).

The Tomita score was used to evaluate the prognosis of patients with spinal metastases [19]. The Frankel score was used to assess the patient’s neurological status [20]. Evaluations of the physical activity of patients were based on ECOS-PS [21]. The best response was evaluated according to the mRECIST for HCC [22], which was classified into complete response (CR), partial response (PR), stable disease (SD), and progressive disease (PD). OS was defined as the time between the delivery of systemic therapy and death. PFS was stated as the time from initiating systemic therapy to documentation of any clinical or radiological disease progression or death, whichever occurred first. We calculated the cumulative risk of the initial SREs by Kaplan–Meier analysis from the time between the operation on the BoM and the first occurrence of SREs after surgery. The results of OS and PFS were all presented in months. The objective response rate (ORR) was defined as the ratio of patients with a complete or partial response after treatment. Disease control rate (DCR) was the ratio of patients with CR, PR and SD. Baseline characteristic variables were compared with Pearson’s Chi-square test or Fisher’s exact test based on the actual. The Kaplan–Meier method and Log-Rank test were used to estimate and compare OS and PFS. All statistical analyses were performed via R studio. A two-sided *p* < 0.05 was considered significant.

Adverse events and corresponding grades were evaluated by the National Cancer Institute’s Common Terminology Criteria for Adverse Events (CTCAE) 5.0 [23]. For patients with serious adverse events (Grade 3–4), symptomatic intervention (medication or physical therapy) should be supplemented in most cases [24,25]. In consideration of those who cannot be alleviated by conservative intervention, appropriate reduction or even withdrawal of systemic therapy is recommended. For instance, for the rash with the highest incidence of grade 3–4 adverse events in this study, symptomatic interventions were topical anti-allergy cream, glucocorticoid cream, dose reduction, etc.

## 3. Results 

### 3.1. Patient Baseline Characteristics

According to the criteria, 62 patients were enrolled in this study: 21 received TKIs alone, and 41 received TKIs + ICIs combination therapy (Table 1). More male patients were included in combination therapy compared with TKIs monotherapy (40/41, 97.6% vs. 17/21, 81.0%; *p* = 0.041). In the combination group, there were 18 (43.9%) and 14 (34.1%) patients with thoracic vertebral metastases and cervical vertebral metastases, respectively, accounting for the vast majority. In the monotherapy group, 5 (23.8%), 5 (23.8%), and 6 (28.6%) patients experienced cervical, thoracic, and lumbar vertebral metastases. Forty (64.5%) patients were infected with the hepatitis B virus (HBV), only one of whom was hepatitis C virus (HCV). In the case of treatment of primary lesions, patterns were similar in monotherapy and combination therapy group (surgery: 16/21, 76.2% vs. 31/41, 75.6%, *p* = 1.00; chemotherapy: 7/21, 33.3% vs. 14/41, 34.1%; *p* = 0.95; radiotherapy: 8/21, 38.1% vs. 16/41, 39.0%; *p* = 0.94). For preoperative assessment, there were 19 (90.5%) and 36 (87.8%) patients with Tomita score of 5–6 in the monotherapy and combination group (*p* = 1.00); 20 (95.2%) and 34 (82.9%) patients with Frankel grade of D in the corresponding groups (*p* = 0.25). More patients in the combination group used BMAs for long periods (10/21, 47.6% vs. 31/41, 75.6%; *p* = 0.028) but without a significant difference in the BMA types between them (*p* = 0.206).

### 3.2. Outcomes of TKIs and TKIs Plus ICIs

The therapeutic response of TKIs monotherapy and TKIs + ICIs combination therapy is shown in Table 2. A higher ORR (1/21, 4.7% vs. 9/41, 22.0%; *p* = 0.14) and DCR (8/21, 38.1% vs.21/41, 51.2%; *p* = 0.37) was observed in the combination therapy than the monotherapy. There were 7 (33.3%) and 12 (29.3%) patients with SD in the monotherapy and combination therapy (*p* = 0.74), while more patients with PR in TKIs plus ICIs group than TKIs group (9/41, 22.0% vs. 0, 0%; =0.022; Figure 1A,B).

Additionally, TKIs + ICIs prolonged the overall survival (18 months vs. 31 months, HR: 0.45, 95%CI 0.20–0.99; *p* = 0.015) and delayed the tumor progression (10 months vs. 23 months, HR: 0.45, 95%CI 0.20–0.99; *p* = 0.014). This benefit was more remarkable in patients with HBV/HCV (OS: 10 months vs. 33 months, HR: 0.24, 95%CI 0.08–0.69, *p* < 0.001; PFS: 8.5 months vs. 33 months, HR:0.24, 95%CI 0,08–0.68; *p* ≤ 0.001; Figure 1C,D) but not statistically significant in non-infected patients (Appendix A Appendix A).

Information on adverse events could not be collected in some patients due to the loss of follow-up. Therefore, this analysis included 15 patients in the monotherapy group and 34 in the combination therapy group (Table 3). For patients who received TKIs alone, the most common adverse events (any grade) were nausea (6/15, 40%), decreased appetite (6/15, 40%), fatigue (5/15, 33.3%), and increased transaminase (4/15, 26.6%); while nausea (15/34, 44.2%), decreased appetite (11/34, 32.3%) and rash (9/34, 26.5%) were most common in TKIs + ICIs group. It is worth noting that the incidence of grade 3–4 rash was higher in the combination group but without significance. None of the patients died of severe adverse events.

### 3.3. Outcomes of BMAs as Adjuvant Therapy in Systemic Regimens

In the perioperative/no use and the long-term use of BMAs groups, 10 (47.6%) and 31 patients (75.6%) received TKIs + ICIs, respectively (*p* = 0.046). Improved survival was found in patients with long-term use of BMAs compared with perioperative or without BMAs (OS: 33 months vs. 16 months, HR: 2.51, 95%CI 1.14–5.52, *p* < 0.005; PFS: 33 months vs. 11 months, HR: 2.60, 95%CI 1.17–5.77, *p* < 0.005; Figure 2A,B). Patients with prolonged use also showed higher ORR (2/21, 9.5% vs. 8/41, 19.5%; *p* = 0.47) and DCR (5/21, 23.8% vs. 24/41, 58.5%; *p* = 0.015) (Table 4), which was ascribed to fewer patients with PD (16/21, 76.2% vs. 17/41, 41.5%; *p* = 0.015) and more with SD (3/21, 14.3% vs. 16/41, 39.0%; *p* = 0.079). Similar outcomes were found between patients receiving denosumab and ZA (Figure 2C,D). However, it could not rule out the possibility that the ratio of patients who received TKIs + ICIs was higher in the long-term use group (Table 1). 

In Table 4, fewer patients with PD (17/26, 65.4% vs. 12/32, 37.5%; *p* < 0.001) and a higher DCR (9/26, 34.6% vs. 20/41, 62.5%; *p* = 0.79) were observed in the denosumab group than ZA group. From Appendix A, there was a trend of lower risk of the initial SREs after surgery in the TKIs + ICIs (37 months vs. 18 months, HR: 0.78, 95%CI 0.36–1.68, *p* = 0.35) and the denosumab group (29 months vs. 24 months, HR: 0.72, 95%CI 0.35–1.49, *p* = 0.35).

## 4. Discussion

This study showed that TKIs plus ICIs offered favorable outcomes compared with TKIs alone to patients with HCC and BoM, especially those with the infection of HBV or HCV. However, similar survival was observed in uninfected patients undergoing the two regimens. Additionally, patients receiving long-term use of BMAs had improved survival than those with perioperative use or no use of BMAs.

Though previously Sorafenib or Lenvatinib alone has been recommended as first-line systemic therapy [1], current studies have demonstrated that combining TKIs and ICIs like pembrolizumab and lenvatinib, or atezolizumab and bevacizumab, yielded better prognosis and became the preferred choice for advanced HCC [7,8,9]. However, recently, LEAP-002, a phase 3 trial comparing outcomes of Lenvatinib plus Pembrolizumab with lenvatinib alone, failed to show significant advantages in OS and PFS in a cohort of HCC patients with mainly non-HBV/HCV causes. However, in IMBRAVE-150, where the cohort was constituted of patients with viral hepatitis, more favorable outcomes were found in combination therapy than in Lenvatinib alone. Similarly, the COSMIC-312 cohort was dominated by patients with non-viral liver cancer, in which the combination therapy did not achieve the expected survival benefits. A subgroup analysis of several large clinical trials also indicated improved survival for patients with virus-induced liver cancer treated with immunotherapy. In contrast, the benefit for patients with non-viral liver cancer was not significant [26].

Hepatitis B virus (HBV) infection is the dominating cause of hepatitis, liver cirrhosis, and HCC in Asia [27]. Over 80% of Chinese HCC patients are infected with HBV. The infection facilitates the genetic mutation of liver cells [28]; moreover, the immune microenvironment of the liver changes, which leads to immune tolerance and the escape of immune surveillance of mutant cells and subsequently accelerated carcinogenesis [29]. Compared to non-viral liver cancer, PD-1 is upregulated in HBV-related patients, which displays a more substantial effect on the immunosuppression but indicates the clinical practice of ICIs. According to a literature review, the TKIs + ICIs combination therapy prolonged PFS of HBV (+) HCC patients compared with those receiving TKIs monotherapy, but there were no statistical differences in HBV (-) patients [30], which was also consistent with the results of our study. In the future design of immunotherapy trials for HCC and clinical applications, stratifications by etiology may replace regional differences [8].

As patients with HCC had superior survival due to benefits from the development of multimodality therapy, a higher incidence of BoM has been observed [3]. When HCC patients suffer from BoM, the infiltrating cells around the lesion destroy the sclerotin and produce PD-L1 and Chemokine ligand 2 (CCL2), also causing bone pain while promoting osteoclast differentiation simultaneously. Previous research has validated that the anti-PD-1 ICIs can prevent the differentiation of osteoclast precursor cells to osteoclasts by intercepting CCL2 generation, which contributes to enhanced anti-tumor immunity [31]. Pre-existing evidence has confirmed the positive effect of TKIs plus ICIs for patients with solid tumors and metastases [32,33,34]. However, limited clinical investigations have focused on the efficacy and safety of immunotargeted therapy and BMAs for metastatic HCC patients with BoM. Our study indicated that the combined regimens offered survival benefits for those patients without severe adverse events.

The FDA has authorized the use of BMAs, including bisphosphonates (ZA and pamidronate) and denosumab (a monoclonal antibody anti-RANKL), for treating patients with all solid tumors and BoM [35,36]. The occurrence of BoM results in a “cold” tumor microenvironment [37], including fewer tumor infiltration lymphocytes (especially cytotoxic T cells), osteoclastogenesis and release of pro-tumor growth factors that would form a loop between immunosuppressive cells and BoM cells [38], which renders tumors less responsive to PD-1 immunotherapy [39] and consequently worse outcomes [40]. BMAs are widely used to inhibit osteoclasts-mediated bone resorption, which may break the “vicious cycle,” ameliorating the immunosuppressive tumor microenvironment. The mechanism of ZA against tumor cells includes the restraint of osteoclasts’ differentiation by repressing the receptor activator of the RANKL/RANK pathway [41]. Denosumab, a human monoclonal antibody against RANKL, plays a role with a similar approach. RANKL/RANK not only participates in osteoclastogenesis but also affects immunity. Previous studies have demonstrated that this pathway is vital in the injury, proliferation, and metastasis of liver cancer cells [41,42,43,44]. Therefore, the BMAs may possess a potential antitumor effect against the primary lesion.

Preclinical studies have shown that BMAs applied as an adjuvant treatment for HCC with BoM could effectively relieve severe bone pain, suppress bone lesions, and was correlated with favorable outcomes [45,46,47]. Albeit with the evidence in preclinical studies, there was a paucity of investigations to validate the synergy of denosumab, a potential potent inhibitor of RANKL/RANK, with TKIs and ICIs despite the development of treatment strategies of HCC and BoM. Theoretically, combining ICIs and RANKL inhibitors may convert the immunosuppressive bone microenvironment to a “hot” one, thus enhancing clinical responses in patients with advanced-stage cancers. The synergistic effect of BMA s+ ICIs has been reported by preclinical studies in other solid tumors (PC, NSCLC, and melanoma) with BoM [11,12,48], which demonstrated survival benefits and delay of primary tumor progression. In our study, the efficacy of BMAs with immunotargeted therapy and two types of BMAs in patients with HCC and BoM have been comprehensively evaluated. It was implied that the survival profit with the addition of BMAs observed in the evidence mentioned earlier may derive from the suppression of osteoclast activity, which may prevent the release of pro-tumor growth factors and consequently, anti-tumorigenic effects of the immune cells could be enhanced. Hence, with more infiltrations of cytotoxic lymphocytes into BoM, a better response of ICIs could be observed.

There is evidence that regardless of the BoM burden of NSCLC patients, the longer time ICIs and denosumab were used concurrently, the more benefit could be achieved [12,48,49]. In clinical practice, patients with BoM may not receive sequential BMAs, especially patients with advanced stages and poor physical condition. Nevertheless, it may not prevent progressions of BoM, which could result in deterioration of quality of life and worse outcomes. Additionally, the effects of the period of BMAs on survival have rarely been investigated. In our study, we compared the outcomes of patients receiving perioperative or no use with long-term use of BMAs combined with systemic therapy. The results demonstrated that continuous administrations of BMAs with systemic therapy contributed to statistically significant survival benefits and a trend to delay the occurrence of SREs—this will be beneficial for the clinical practice of BMAs plus immunotargeted therapy for patients with HCC and BoM.

Although this pilot study evaluated the efficacy of BMAs combined with systematic therapy, the major limitation of this study is its retrospective nature. Secondly, the sample is small, which may be due to the fact that the number of patients with BoM undergoing sequential systemic therapy is limited. In Table 1, there is a statistical difference in the sex ratio between the TKI mono-therapy group and the combination therapy group—because the incidence of HCC is much higher in men than in women, this resulted in a smaller sample size in women [50]. Therefore, the result of this study is inclined to reflect the systemic therapy of male patients with HCC and BoM. Finally, as a real-world clinical study, long-term follow-up is required to obtain more treatment data, such as PD-1 or PD-L1 intensity of tumor tissues and follow-up imaging data to assess responses of patients with BoM to treatment, which may be conducive to identifing biomarkers of BMAs plus immunotargeted therapy for patients with HCC and BoM. Therefore, these results in our study still need to be validated in prospective clinical trials and more extensive sample-size studies for female patients.

## 5. Conclusions

The combination therapy TKIs + ICIs, compared with TKIs alone for HCC with BoM, achieved favorable outcomes, and without severe toxicities. This advantage was more pronounced in cohorts of patients due to viral causes by subgroup analysis. Systemic therapy combined with long-term use of BMAs has shown remarkable disease remission in the patients. Compared with ZA, denosumab has an advantage in delaying the progression of tumor lesions, but without statistical difference.

## Figures and Tables

**Figure 1 jcm-11-06901-f001:**
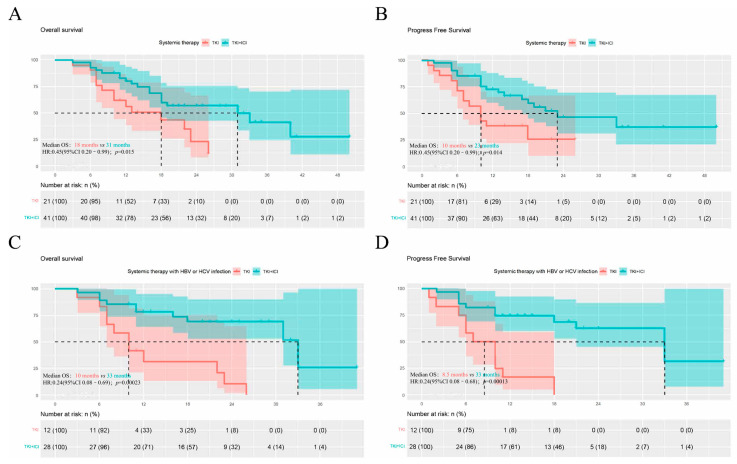
Kaplan-Meier curve of overall and progression-free survival (**A**) Overall survival for TKIs mono-therapy and TKIs plus ICIs groups (**B**) Progression-free survival for TKIs mono-therapy and TKIs plus ICIs groups (**C**) Overall survival for TKIs mono-therapy and TKIs plus ICIs groups with HBV/HCV infection (**D**) Progression-free survival for TKIs mono-therapy and TKIs plus ICIs groups with HBV/HCV infection.

**Figure 2 jcm-11-06901-f002:**
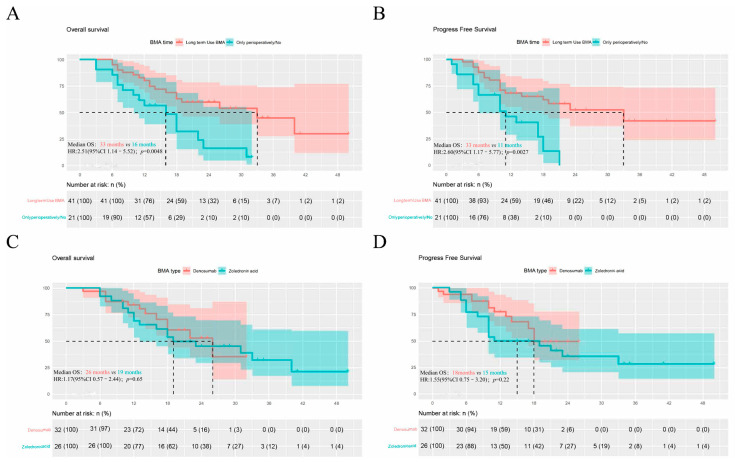
Kaplan–Meier curve of overall and progression-free survival: (**A**) Overall survival for using BMA in the long term and only using it perioperatively, or not at all; (**B**) Progression-free survival for using BMA in the long term and only using it perioperatively, or not at all; (**C**) Overall survival for ZA and denosumab groups; and (**D**) Progression-free survival for ZA and denosumab groups.

**Table 1 jcm-11-06901-t001:** Baseline characteristics of patients receiving TKIs monotreatment and TKIs + ICIs combination therapy, N = 62 (%).

	TKIs Monotreatment, N = 21	TKIs + ICIs Combination Therapy, N = 41	Total, N = 62	*p* Value
**General Information**				
Age:				0.15
Mean (SD)	57.5 (11.2)	53.3 (9.70)	54.7 (10.3)	
Median [IQR: Q1-Q3]	56.0 [47.0–62.25]	53.0 [46.5–58.5]	53.0 [47.0–61.75]	
Gender:				0.041
Male/Female	17 (81.0%)/4 (19.0%)	40 (97.6%)/1 (2.4%)	57 (91.9%)/5 (8.1%)	
ECOG-PS:				0.58
0–1/≥2	15 (71.4%)/6 (28.6%)	25 (61.0%)/16 (39.0%)	40 (64.5%)/22 (35.5%)	
**Preoperational Data**				
Metastatic bone location:				0.090
Cervical/Thoracic/Lumber Sacrum/Limb bone	5 (23.8%)/5 (23.8%)/ 6 (28.6%) 4 (19.0%)/1 (4.8%)	14 (34.1%)/18 (43.9%)/ 7 (17.1%) 2 (4.9%)/0 (0%)	19 (30.6%)/23 (37.1%)/ 13 (21.0%) 6 (9.7%)/1 (1.6%)	
HBV/HCV infection:				0.41
Yes/No	12 (57.1%)/9 (42.9%)	28 (68.3%)/13 (31.7%)	40 (64.5%)/22 (35.5%)	
Child-Pugh:				0.14
A/B	20 (95.2%)/1 (4.8%)	32 (78.0%)/9 (22.0%)	52 (83.9%)/10 (16.1%)	
Surgery on the primary lesion:				1.00
Yes/No	16 (76.2%)/5 (23.8%)	31 (75.6%)/10 (24.4%)	47 (75.8%)/15 (24.2%)	
AFP:				0.23
<400 ng/mL/≥400 ng/mL	18 (85.7%)/3 (14.3%)	29 (70.7%)/12 (29.3%)	47 (75.8%)/15 (24.2%)	
Preoperative Frankel grade:				0.76
A/B	0 (0%)/0 (0%)	2 (4.9%)/2 (4.9%)	2 (3.2%)/2 (3.2%)	
C/D	1 (4.8%)/20 (95.2%)	3 (7.3%)/34 (82.9%)	4 (6.5%)/54 (87.1%)	
Preoperative Tomita score:				1.00
5–6/7–8	19 (90.5%)/2 (9.5%)	36 (87.8%)/5 (12.2%)	55 (88.7%)/7 (11.3%)	
**Sequential Treatment**				
Visceral Metastases:				0.90
None/Lung/Multiple	13 (61.9%)/2 (9.5%)/ 1 (4.8%)	26 (63.4%)/4 (9.8%)/ 4 (9.8%)	39 (62.9%)/6 (9.7%)/ 5 (8.1%)	
Intrahepatic metastasis/Other locations	5 (23.8%)/0 (0%)	6 (14.6%)/1 (2.4%)	11 (17.7%)/1 (1.6%)	
Chemotherapy:				1.00
Yes/No	7 (33.3%)/14 (66.7%)	14 (34.1%)/27 (65.9%)	21 (33.9%)/41 (66.1%)	
Radiotherapy:				1.00
Yes/No	8 (38.1%)/13 (61.9%)	16 (39.0%)/25 (61.0%)	24 (38.7%)/38 (61.3%)	
The type of BMAs:				0.21
No/Zoledronic acid	3 (14.3%)/8 (38.1%)	1 (2.4%)/18 (43.9%)	4 (6.5%)/26 (41.9%)	
Denosumab	10 (47.6%)	22 (53.7%)	32 (51.6%)	
The time of BMAs use:				
Long term use/Only perioperatively or No	10 (47.6%)/11 (52.4%)	31 (75.6%)/10 (24.4%)	41 (66.1%)/21 (33.9%)	0.046

Abbreviations: *TKIs* tyrosine kinase inhibitors, *ICIs* immune-checkpoint inhibitors; *ECOG-PS* Eastern Cooperative Oncology Group-performance status; *BMAs* bone-modifying agents.

**Table 2 jcm-11-06901-t002:** Best response of TKIs and TKIs + ICIs, N = 62 (%).

	TKIs, N = 21	TKIs + ICIs, N = 41	Total, N = 62	*p* Value
mRECIST:				
Complete response	1 (4.7%)	0 (0%)	1 (1.6%)	0.34
Partial response	0 (0%)	9 (22.0%)	9 (14.5%)	0.022
Stable disease	7 (33.3%)	12 (29.3%)	21 (33.9%)	0.78
Progressive disease	13 (61.9%)	20 (48.8%)	33 (53.2%)	0.42
Objective Response Rate (ORR)	1 (4.7%)	9 (22.0%)	10 (16.1%)	0.14
Disease Control Rate (DCR)	8 (38.1%)	21 (51.2%)	29 (46.8%)	0.42

Abbreviations: *mRECIST* modified Response Evaluation Criteria in Solid Tumors.

**Table 3 jcm-11-06901-t003:** Systemic therapy-related adverse events, N = 49 (%).

	TKIs Mono-Treatment, N = 15		TKIs + ICIs Combination, N = 34		*p* Value	
	Any Grade	Grade 3–4	Any Grade	Grade 3–4	Any Grade	Grade 3–4
Nausea	6 (40.0%)	0 (0%)	15 (44.1%)	0 (0%)	1.00	1.00
Fatigue	5 (33.3%)	0 (0%)	6 (17.6%)	0 (0%)	0.28	1.00
Pruritus	2 (13.3%)	0 (0%)	6 (17.6%)	1 (2.9%)	1.00	1.00
Rash	3 (20.0%)	0 (0%)	9 (26.5%)	6 (17.6%)	0.73	0.16
Myasthenia	2 (13.3%)	0 (0%)	3 (8.8%)	0 (0%)	0.64	1.00
Decreased Appetite	6 (40.0%)	1 (6.67%)	11 (32.4%)	0 (0%)	0.75	0.31
Colitis	1 (6.7%)	0 (0%)	2 (5.9%)	0 (0%)	1.00	1.00
Hemorrhagic Tendency	1 (6.7%)	0 (0%)	2 (5.9%)	0 (0%)	1.00	1.00
Dyspnea	0 (0%)	0 (0%)	2 (5.9%)	0 (0%)	1.00	1.00
Pyrexia	1 (6.7%)	0 (0%)	2 (5.9%)	0 (0%)	1.00	1.00
Increased Transaminase	4 (26.7%)	2 (13.3%)	2 (5.9%)	1 (2.9%)	0.062	0.22
Hypertension	0 (0%)	0 (0%)	3 (8.8%)	0 (0%)	0.54	1.00
Hand-foot syndrome	2 (13.3%)	0 (0%)	2 (5.9%)	0 (0%)	0.58	1.00
Hypothyroidism	0 (0%)	0 (0%)	2 (5.9%)	1 (2.9%)	1.00	1.00
Hypocalcemia	1 (6.7%)	0 (0%)	5 (14.7%)	0 (0%)	0.65	1.00
Vitiligo	0 (0%)	0 (0%)	1 (2.9%)	0 (0%)	1.00	1.00

**Table 4 jcm-11-06901-t004:** Best response of BMA-related treatment, N = 62 (%).

	Only Used BMAs Perioperatively/No Treatment (N = 21)	Long Term Use of BMAs (N = 41)	Total (N = 62)	*p* Value
mRECIST:				
Complete response	1 (4.7%)	0 (0%)	1 (1.6%)	0.34
Partial response	1 (4.7%)	8 (19.5%)	9 (14.5%)	0.15
Stable disease	3 (14.3%)	16 (39.0%)	19 (30.6%)	0.079
Progressive disease	16 (76.2%)	17 (41.5%)	33 (53.2%)	0.015
Objective Response Rate	2 (9.5%)	8 (19.5%)	10 (16.1%)	0.47
Disease Control Rate	5 (23.8%)	24 (58.5%)	29 (46.8%)	0.015

## Data Availability

The data presented in this study are available on request from the corresponding author.

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
