# Peer review of "Combination of Bone-Modifying Agents with Immunotarget Therapy for Hepatocellular Carcinoma with Bone Metastases"

_jcm, 2022, doi:10.3390/jcm11236901_

Round 1
Reviewer 1 Report
It is valuable to compare the clinical outcome of TKI with TKI+ICI, as well as the impact of BMA in bone metastasis of HCC. This work enrolled 62 patients and studied the clinical outcome and adverse events of these treatments. Overall, it is important and significant to this field.
(1) It is better to provide the ethics approved No. in the Method section.
(2) It is better to describe how to manage the grade 3-4 adverse events in the Method section.
(3) In this study, the number of male patients are more than female. It is better to discuss whether this could impact the conclusion of this study in the Discussion section.
(4)Table 3, it is better to clearly seperate the display of the text of these two groups "TKIs Mono-treatment,N=15" and "TKIs+ICIs Combination, N=34". Such as using broken line under these text?
Author Response
Please see the attachment. We sincerely appreciate your review and comments.

Reviewer 2 Report
This manuscript is topical and interesting. Immunotarget therapy for hepatocellular carcinoma with metastases is certainly an important clinically relevant area and ripe for expansion. While the clinical data and hypothesis look robust. There are several issues to address before the manuscript can move forward:
- The abstract needs to be revised with the sub-headings removed.
- The introduction would benefit from an additional context of kinase inhibitors in cancer, there are alot of key references missing that would give this work a better grounding. eg https://pubmed.ncbi.nlm.nih.gov/34354255/, https://pubmed.ncbi.nlm.nih.gov/33513356/, https://pubmed.ncbi.nlm.nih.gov/34921994/
- There is also an English issue throughout the paper that needs addressing eg 3rd paragraph second sentence is very long, 5 lines, 5 commas and 2 'and's. There are other issues including sentences starting with 'And...' eg page 3 line 7 and 13.
- Tables should be formatted to be on one page.
- Figure quality needs to significantly improved, the current versions are too small and low resolution.
- The referencing in this paper is very thin eg page 8 - The FDA has authorized the use of BMAs, including bisphosphonates (ZA and pamidronate)and denosumab (a monoclonal antibody anti-RANKL), for treating patients with all solid tumors and BoM. - no reference? This despite ZA being a central benchmark in the narrative.
- It is not typical to have 4 co-first authors, this is an internal discussion, but as a reader it seems abit odd.
Author Response
Responses to reviewer #2’s comments:
Point (1): The abstract needs to be revised with the sub-headings removed.
Our response 1: We apologize for these flaws of abstract format. Sub-headings have been removed.
Point (2): The introduction would benefit from an additional context of kinase inhibitors in cancer, there are alot of key references missing that would give this work a better grounding. eg https://pubmed.ncbi.nlm.nih.gov/34354255/, https://pubmed.ncbi.nlm.nih.gov/33513356/, https://pubmed.ncbi.nlm.nih.gov/34921994/
Our response 2: Thank you very much for offering key as well as meaningful articles. The references on kinase inhibitors recommended have been cited.
Point (3): There is also an English issue throughout the paper that needs addressing eg 3rd paragraph second sentence is very long, 5 lines, 5 commas and 2 'and's. There are other issues including sentences starting with 'And...' eg page 3 line 7 and 13.
Our response 3: We believe this is a very precious suggestion which may make our language more normalized. The sentences beginning with “And” have been modified, and the length of some sentences has been shortened.
Point (4): Tables should be formatted to be on one page.
Our response 4: We are very agreeable to the comment on formatting adjustment of Table1. The table has been adjusted to one page. But the format is aesthetically limited; if you have a better suggestion for the form format, please do not hesitate to contact me. Revision mode is not enabled for changes to the table because we want to render the format adjusted for your reviewing.
Point (5): Figure quality needs to significantly improved, the current versions are too small and low resolution.
Our response 5: We apologize for the quality defects of the figures -- this may be due to the fact that the software compressed the figures when adjusting the format in the pre-layout. Figures 1 and 2 have been replaced with clearer versions.
Point (6): The referencing in this paper is very thin eg page 8 - The FDA has authorized the use of BMAs, including bisphosphonates (ZA and pamidronate)and denosumab (a monoclonal antibody anti-RANKL), for treating patients with all solid tumors and BoM. - no reference? This despite ZA being a central benchmark in the narrative.
Our response 6: We apologize for this omission of reference. The paper on ZA and denosumab at the time of their FDA approval was found and added to the discussion. We appreciate your precise as well as thorough review of the discussion.
Point (7): It is not typical to have 4 co-first authors, this is an internal discussion, but as a reader it seems abit odd.
Our response 7: Thanks for the comment. The contributions of the 4 co-first authors in this study are all irreplaceable. It is not easy to collect and manage long-term treatment data of patients with HCC and BoM because the survival period of the patients is relatively short after diagnosis, and the condition is often complex. P. W. is an experienced clinician of orthopedic tumor, and Z.S., an M.D. student, has invested much energy in this study. This study is a single-center clinical study of the largest sample size of HCC with BoM so far; we hope readers can understand the "four Co-first authors" based on this hard-won data. X. W. has made irreplaceable contributions to the conceptual shaping and scheme planning of this study. In the subsequent data processing and statistical operation, Z S. and Z C. (the writer of this manuscript) undertook the main work. We were going to consider P W. as one of the correspondents, but then there would be three corresponding authors, which is also a bit odd for a single-center study. The specific contributions of each author are also attached at the end of the text. Given the above considerations, we have decided to retain the current authorship arrangement. We sincerely hope to get your understanding.

Round 2
Reviewer 2 Report
-